# Parent values and preferences underpinning treatment decision making in poor prognosis childhood cancer: a scoping review protocol

Helen Pearson  ,[1,2] Faith Gibson,[3,4] Anne-Sophie Emma Darlington[5]

[1]Royal Marsden NHS Foundation Trust, London, UK
[2]Faculty of Medicine Health and Life Sciences, University of Southampton, Southampton, UK
[3]Faculty of Health and Medical Sciences, University of Surrey, Guildford, UK
[4]Great Ormond Street Hospital For Children NHS Foundation Trust, London, UK
[5]Faculty of Health Sciences, University of Southamptom, Southampton, UK

**Correspondence to**
Helen Pearson;
helenpearson1@nhs.net

## ABSTRACT

**Introduction** Parents of a child with cancer want to be involved in making treatment decisions for their child. Underpinning and informing these decisions are parents' individual values and preferences. Parents of a child who has a poor prognosis cancer and who subsequently dies can experience decisional regret. To support parents, and potentially reduce decisional regret, identifying the values and preferences of parents who are making these treatment decisions may enhance the support that can be provided by healthcare professionals. An increased understanding will support future work in this area and identify research gaps that could strengthen support strategies in clinical practice. The aim of this scoping review is to explore parent values and preferences underpinning treatment decision making when their child is receiving cancer-directed therapy for a poor prognosis cancer.

**Methods and analysis** The Joanna Briggs Institute scoping review methodology will be followed. An initial database search of CINHAL and MEDLINE will be conducted to analyse the keywords using subject headings and Medical Subject Headings terms. Articles will be initially screened on title and abstract. The reference and citation lists of the full-text articles to be included will be searched using Web of Science. Articles will be independently reviewed by two reviewers and any discrepancies discussed with a third reviewer. Data extracted will be presented in tabular, diagrams and descriptive summaries.

**Ethics and dissemination** Ethical approval is not required for this scoping review. This review will inform further research with parents to understand their values and preferences when making repeated treatment decisions when their child has a poor prognosis cancer. All outputs will be disseminated through peer-reviewed publications and conference presentations.
This scoping review is registered on the Open Science Framework (https://osf.io/n7j9f).

## Strengths and limitations of this study

► This is the first scoping review to identify parent values and preferences in treatment decision making when their child has a poor prognosis cancer.

► The search strategy includes the reference and citation lists of all included full-text articles for a comprehensive search and cross-checking purposes.

► The Joanna Briggs Institute methodology provides a systematic approach to undertaking and reporting this scoping review.

► Grey literature will not be included due to the niche research field and extensive international research collaboration resulting in all available literature being published.

► There is no formal quality assessment within a scoping review. A narrative synthesis of the research literature reviewed will be undertaken and published.

## INTRODUCTION

Globally, approximately 300 000 children and young people from birth to 19 years of age are diagnosed with cancer each year[1] and 90 000 of these children die from the disease.[2] Within the UK, of the 1420 children under the age of 15 years diagnosed with cancer every year,[3] roughly 250 children will die.[4] Parent involvement in their child's care can include making decisions regarding treatments, administering medications and advocating for their child within the multidisciplinary team. Research has shown that parents want to be involved in all aspects of their child's care particularly when it comes to making treatment decisions.[5] However, parents are rarely involved in decision making with initial treatment as this is protocol driven, with reduced choices, but take a more active role if treatments fail and relapse strategies are required.[6]

Studies investigating parental experiences of having a child with cancer have tended to focus on diagnoses such as leukaemia or lymphoma, which generally have a good prognosis.[3] Parents of children with a poor prognosis cancer are under-represented. The needs and experiences of this parent population are likely to be different because: (1) survival rates are lower[3]; (2) the number of children diagnosed with a cancer that has a

poor prognosis is lower[3]; and (3) established treatment protocols are not always available due to low numbers of children being diagnosed, resulting in a longer time for clinical trials to become established.

Clinical trials within paediatric oncology have advanced over the past decade resulting in more treatment options for children with a poor prognosis cancer. There has been a vast number of clinical trials involving immunotherapy and targeted inhibitors that have shown to improve or extend survival and in some cases provide cures within adult cancer.[7 8] As a result, these have been extended into clinical trials for children particularly for those who have a poor prognosis cancer.[9 10] Reflections on clinical practice have identified that where established treatment protocols may not be available; parents subsequently become involved in treatment decision making. This includes options such as clinical trials and off-protocol treatments. Often parents can make a series of repeated treatment decisions depending on their child's response to treatment and toxicities experienced. This is the case for parents whose child is diagnosed with relapsed neuroblastoma or relapsed medulloblastoma.

Prognosis relates to the outcome of the cancer diagnosis and whether the disease is likely to return.[11] A poor prognosis indicates a highly unfavourable treatment outcome likely to result in death.[11] Emerging research conducted by Lichtenthal *et al*[12] published in 2020 has shown that 73% of parents whose child has a poor prognosis and subsequently dies experience regret. Regret is predominately related to the decisions parents have made, and this regret can result in prolonged grieving.[13]

Research published in 2011 by Tomlinson *et al*[14] has acknowledged that parents pursue cancer-directed treatments even when the chance of cure is small. Although this research is a decade old, it remains a pertinent piece of research supporting what is seen in clinical practice today. This suggests parents continue to pursue cancer-directed treatments regardless of cure outcomes. Quality information including treatment options, risks and benefits, expectations of treatment and cure to make informed decisions based on parent individual values and preferences is paramount.[5] This is important in treatment decision making to increase parent empowerment, confidence in the decision made[15] and potentially reduce decisional conflict and regret.[12] Quality information relating to diagnosis, treatment, expectations of cure, potential future impairments, how their child is responding to treatment and how the cancer may have developed are important factors for parents.[16] Research investigating parent information needs has highlighted that good quality information results in higher levels of satisfaction in healthcare professionals and the care their child is receiving.[16 17] Research on parent treatment decision making at the end of life or opting to participate in phase I clinical trials has shown parent preferences for cancer-directed treatment includes hoping for a cure, prolonging of life, decreasing suffering[5 14] and ensuring they have done everything possible to save their child's life.[14] With the recognition

that parents experience high levels of regret, and want to be involved in treatment decision making, particularly when their child has a poor prognosis, this requires an understanding of individual parent values and preferences that inform these decisions.

Values and preferences relate to the expectations, goals, beliefs and predispositions of an individual when making a decision.[18] These values and preferences inform the decision-making process for a person and ultimately define what decision they make. Supporting parents in understanding their own values and preferences and how these inform treatment decision making is essential to help reduce regret. Support tools to elicit parent values and preferences can help in the process of making decisions.[19] As regret in this parent population is centred on treatment decision making, understanding parent values and preferences may provide insight into what parents require in terms of support to enable them to make informed decisions.

This scoping review has been prompted by the author's clinical experience of caring for children receiving cancer-directed therapy for a poor prognosis cancer. Parents are involved in making treatment decisions for their child, often with the need to make a series of repeated treatment decisions as their child's condition changes. This raises questions such as: what are parent values and preferences when their child is receiving cancer-directed therapy for a poor-prognosis cancer? How do these values and preferences inform treatment decision making?

A preliminary search on MEDLINE using the EBSCO platform in April 2020 found two relevant reviews. A literature review published in 2013 investigated parent decision making of care and treatment in childhood cancer.[20] This review did not specifically focus on parent values and preferences when their child has a poor prognosis cancer where decision making may differ due to the child's condition and likely outcome. It cannot be assumed that values and preferences would be the same for parents regardless of the outcome for their child (survival, death and disabilities as a result of diagnosis/treatment). This warrants specific investigation. A mixed-methods systematic review published in 2014 looked at medicolegal practice in judgements and decisions when a child's treatment is not curative.[21] The outcome from this review provides a comprehensive theoretical framework on the processes of parent decision making. The theoretical framework is of the positive and negative influences in the decision-making process focusing on the transition from cancer-directed therapy to palliative/end-of-life care for their child.[21] How these decisions relate to individual values and preferences that underpins parents' decision making is not included in this framework. The framework also considers the illness trajectory from cancer-directed therapy to palliative/end-of-life care to be a linear process, which in clinical practice is not always the case.

This scoping review allows for a primary focus on parent values and preferences when their child is receiving cancer-directed therapy for a poor prognosis cancer; this

has not been specifically addressed previously. This review seeks to map the available research literature of parent values and preferences underpinning treatment decision making when their child is receiving cancer-directed therapy for a poor prognosis cancer. Through clarifying these values and preferences, this scoping review will inform further research with parents who are making repeated treatment decisions when their child has a poor prognosis cancer. Data from this scoping review and data from interviews with parents who are making repeated treatment decisions will be combined. This will inform the development of a decision aid that is underpinned by parent values and preferences to support parents when making treatment decisions. Furthermore, the scoping review may identify research gaps that could be addressed in the future to strengthen the support given to parents when making treatment decisions for their child.

## REVIEW OBJECTIVES AND QUESTIONS

The objective is to explore parent values and preferences underpinning treatment decision making when their child is receiving cancer-directed therapy for a poor prognosis cancer. This scoping review will seek to answer two questions:

1. What are parent values and preferences when their child is receiving cancer-directed therapy for a poor prognosis cancer?
2. How do these values and preferences inform treatment decision making?

### Inclusion criteria
#### Participants
This will be parents of a child (<18 years) who has a poor prognosis cancer and is receiving cancer-directed therapy.

### Concept
The aim is to describe, explore and understand the values and preferences that underpin parent treatment decision making when their child is receiving cancer-directed therapy for a poor prognosis cancer. The review will specifically focus on studies where the child is receiving cancer-directed therapy, defined as any type of cancer treatment as opposed to end-of-life care, palliative care or symptom management alone. The parent of a child who is receiving cancer-directed therapy and palliative care/symptom management will be included.

### Context
Parents of a child who is receiving cancer-directed therapy for a poor prognosis cancer in any clinical or medical setting (hospital, community and hospice) will be considered.

### Types of evidence sources
International evidence sources for this review will be considered. As this is a small parent population within childhood cancer, international collaboration to enhance knowledge and progress research within this field is well established. Qualitative, quantitative and mixed methods studies drawn from research literature including systematic reviews will be included. Literature or systematic reviews that report the same study population as research studies included will be considered eligible as long as different data are presented. Otherwise, these articles will be included and used for cross-checking purposes.

Studies included can be prospective or retrospective. Including retrospective studies is important due to the sensitivity of the topic as research has been carried out with parents after their child has died to reduce burden prospectively.

The search will include the last 25 years from 1995 to 2020 inclusive (until the time the database searches take place). This time period has been considered as seminal work was published in 1996 relating to parent treatment decision making when their child has a first cancer recurrence.[22] In addition, being a small parent population, research can take years to complete due to low participant numbers; therefore, an extensive search period is required.

## METHODS

The Joanna Briggs Institute scoping review methodology will be followed.[23] This scoping review was registered on the Open Science Framework on 2 June 2020 (https://osf.io/n7j9f). For the scoping review report, the Preferred Reporting Items for Systematic Reviews and Meta-Analyses (PRISMA-ScR) Checklist[24] will be followed (see online supplemental figue 1). The Preferred Reporting Items for Systematic Reviews and Meta-Analyses flow chart will provide the search process of identification, screening, eligible and included articles for the scoping review (figure 1).

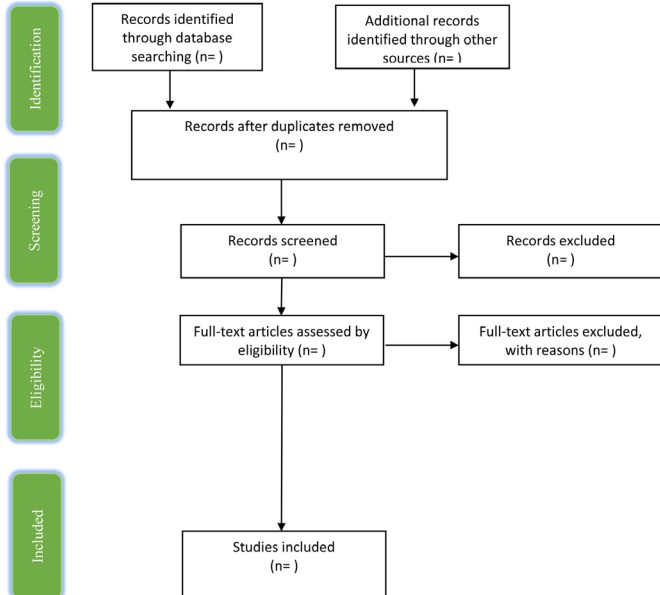

**Figure 1** PRISMA flow chart. PRISMA, Preferred Reporting Items for Systematic Reviews and Meta-Analyses.

**Table 1** Keywords for subject headings/MeSH terms and phrases for database searches

| Keywords searched for subject headings/MeSH terms and relevant phrases | | | | | |
|---|---|---|---|---|---|
| **Parent** | **Value** | **Poor prognosis** | **Child** | **Cancer** | **Decision making** |
| Family | Preference | Relapse | Paediatric | Oncology | "decision making" |
| | Choice | Recurrent | Pediatric | Neoplasm | |
| | Choose | Refractory | | Malignancy | |
| | Belief | Incurable | | "Cancer-direct* therap" | |
| | Attitude | Advanced cancer | | | |
| | Expectation | Uncertainty | | | |
| | Predisposition | Deteriorate | | | |
| | Influence | Disease progression | | | |
| | Experience | "advanc* cancer" | | | |
| | Perception | "no realistic chance of cure" | | | |
| | "goal* of care" | "disease* progress*" | | | |
| | | "palliat* chemotherap*" | | | |
| | | "no reasonable chance of cure" | | | |

MeSH, Medical Subject Headings.

## Search strategy

Database searches will ascertain published research studies, literature or systematic reviews. An initial search was completed in MEDLINE and Cumulative Index of Nursing and Allied Health Literature (CINAHL) using the EBSCO platform in April 2020. This initial database search found relevant primary studies using qualitative and quantitative methods and therefore warrants the scoping review being conducted. The MEDLINE search yielded 216 articles, and the CINAHL search yielded 115 articles. The key words were devised from known terminology used in clinical practice, academia, definitions (written in the introduction) and literature read on parent decision making in childhood cancer. Table 1 shows the list of key words and phrases. Each key word was searched for subject headings/Medical Subject Headings (MeSH) terms to allow for full exploration of the academic language associated with the word. Phrases were not searched for subject headings/MeSH terms as these are natural language phrases as opposed to academic language. The MEDLINE search strategy used in the initial database search is detailed in figures 2–4. From the articles retrieved in this initial search, the text words in the title and abstract were analysed in combination with the index terms used to describe the articles. This did not provide any additional key words or phrases to be included in the database searches.

The databases to be searched for this review will be CINAHL, MEDLINE and PsycINFO using the EBSCO platform and the Web of Science core collections using the Clarivate platform. The list of key words will be searched for subject headings/MeSH terms separately in each of the databases with the acknowledgement that databases do not have a consistent approach to indexing articles. Phrases will not be searched for subject

headings/MeSH terms. The use of search symbols such as truncation, adjacency and the wildcard will allow for full exploration of words. For example, p#ediatric provides inclusion of English and US spelling of the word. Exact phrase searching with the use of quotation marks, for example "decision-making" and "no realistic chance of cure" will be incorporated. This allows for inclusion of some commonly used phrases in clinical practice and within the research literature previously read. Boolean operators (AND/OR) will support the combination between key words, subject headings/MeSH terms using OR and the main topics using AND (parent, values and

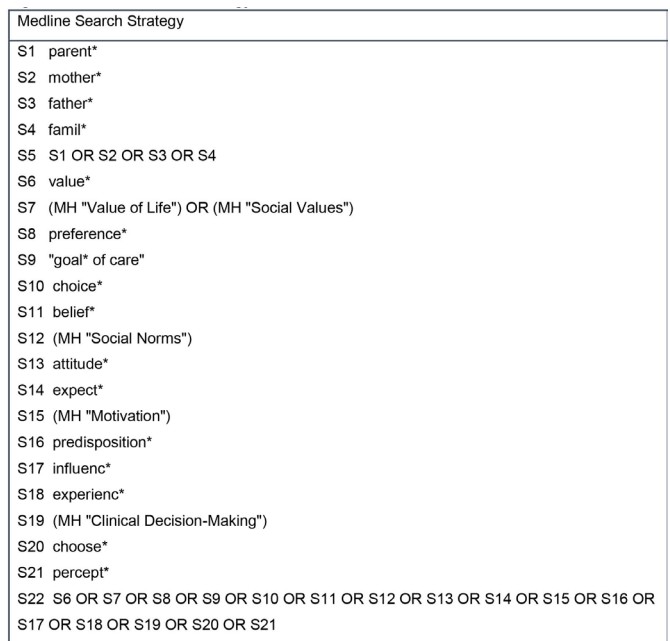

| Medline Search Strategy |
|---|
| S1   parent* |
| S2   mother* |
| S3   father* |
| S4   famil* |
| S5   S1 OR S2 OR S3 OR S4 |
| S6   value* |
| S7   (MH "Value of Life") OR (MH "Social Values") |
| S8   preference* |
| S9   "goal* of care" |
| S10  choice* |
| S11  belief* |
| S12  (MH "Social Norms") |
| S13  attitude* |
| S14  expect* |
| S15  (MH "Motivation") |
| S16  predisposition* |
| S17  influenc* |
| S18  experienc* |
| S19  (MH "Clinical Decision-Making") |
| S20  choose* |
| S21  percept* |
| S22  S6 OR S7 OR S8 OR S9 OR S10 OR S11 OR S12 OR S13 OR S14 OR S15 OR S16 OR S17 OR S18 OR S19 OR S20 OR S21 |

**Figure 2** MEDLINE search strategy.

```
S23  "decision making"
S24  (MH "Decision Making, Shared")
S25  decision N2 making
S26  decision*
S27  decide*
S28  S23 OR S24 OR S25 OR S26 OR S27
S29  S5 AND S22 AND S28
S30  "poor prognosis"
S31  relaps*
S32  (MH "Recurrence")
S33  recurrenc*
S34  refractory
S35  incurable
S36  (MH "Terminally Ill")
S37  "advanc* cancer"
S38  "no realistic chance of cure"
S39  uncertain*
S40  deteriorat*
S41  (MH "Clinical Deterioration")
S42  "disease* progress*"
S43  "palliat* chemotherap*"
S44  "no reasonable chance of cure"
S45  S30 OR S31 OR S32 OR S33 OR S34 OR S35 OR S36 OR S37 OR S38  OR S39 OR
S40 OR S41 OR S42
      OR S43 OR S44
S46  child*
```

**Figure 3**  MEDLINE search strategy.

preferences, decision making, poor prognosis, child and cancer). The main topics combined using AND in the MEDLINE search strategy (figures 2–4) are parent (S5), values and preferences (S22), decision making (S28), poor prognosis (S45), child (S48) and cancer (S56).

Inclusion criteria for the review includes: published research studies, literature or systematic reviews written in English with full abstract available from January 1995 onwards. Articles can be from any country. Only articles written in English will be included due to the feasibility of being able to transcribe research published in other languages and the time/financial cost in doing so. Articles will be initially screened on the title and then refined through reading the abstract before retrieving the full text to confirm inclusion or exclusion. The Web of Science will be used to look at the reference and citation lists of all included full-text articles. This will highlight any additional articles not included in the database searches that may need to be considered for inclusion. Including citation lists is known as snowballing (or pearl growing) and considered an important aspect to ensure inclusivity of all

```
S47  p#ediatric*
S48  S46 OR S47
S49  cancer
S50  neoplasm*
S51  (MH "Neoplasm Recurrence, Local")
S52  malignan*
S53  (MH "Neoplasms")
S54  "cancer direct* therap*"
S55  oncolog*
S56  S49 OR S50 OR S51 OR S52 OR S53 OR S54 OR S55
S57  S45 AND S48 AND S56
S58  S29 AND S57
```

**Figure 4**  MEDLINE search strategy.

relevant articles.[25] Grey literature will not be included as this is a niche topic with established international collaborations resulting in available research and literature reviews being published.

The corresponding author for any article will be contacted if clarification is required for completeness of either including or excluding the article within the scoping review. The database searches will be peer reviewed by the University Research Librarian for transparency and robustness of the search strategy.

### Source of evidence selection

Articles obtained will be stored on EndNote X9.2, and duplications will be removed. Initial screening of the studies will be done by reading the title and abstract to ascertain whether they fit the inclusion criteria. This screening will be carried out independently by two reviewers (HP and A-SED) before the review team meet to discuss the articles to be included for full-text review. Any discrepancies will be resolved with the addition of the third reviewer (FG).

Full text of the articles included based on abstract will be retrieved. These will be read in full by two reviewers (HP and A-SED). Corresponding authors of the retrieved articles will be contacted for clarification if required. In the absence of being able to contact the corresponding author of an article, other members of the research team will be approached. The review team will meet again to discuss the final articles to be included in the scoping review, and the third reviewer (FG) will resolve any discrepancies. Any full-text articles that are excluded at this stage will be justified in the scoping review report. For transparency, the search results will be presented in the PRISMA flow chart (figure 1), which will systematically detail the selection process of articles.

### Data extraction

Data will be extracted from the articles obtained that meet the objectives of the scoping review. Data extraction will be undertaken in full by two reviewers (HP and A-SED), and the third reviewer (FG) will randomly select 25% of the eligible articles to review. The third reviewer (FG) will review data extraction templates completed by the two full reviewers (HP and A-SED) against the relevant article. This is a quality check to ensure the extracted data from the articles are accurate and are executed with rigour. Once the data extraction and review of the randomly selection articles have been completed independently by all three reviewers, the review team will meet to discuss and address any discrepancies.

For a consistent approach between reviewers, the Joanna Briggs Institute source of evidence template[23] will be modified to extract information, characteristics and findings from the articles that are relevant to answering the scoping review questions. This will include information on the population, concept, context, type of evidence source, study methods, values and preferences, information on what cancer-directed therapy was given

| Scoping Review Details | |
| --- | --- |
| *Scoping Review Title* | |
| *Review Objectives* | |
| *Review Questions* | |
| Inclusion/Exclusion Criteria | |
| *Population (parent of a child under age of 18 years)* | |
| *Concept (parent of a child receiving cancer-directed therapy for a poor prognosis cancer)* | |
| *Context (hospital, community, hospice)* | |
| *Types of Evidence Source (Qual/Quant, mixed methods)* | |
| Evidence source Details and Characteristics | |
| *Citation details (author, year of publication, journal)* | |
| *Country* | |
| *Study Aims/Objectives* | |

**Figure 5** JBI data extraction tool. JBI, Joanna Briggs Institute.

and whether this was in conjunction with palliative care/symptom management. The modified data extraction template for this scoping review can be found in figures 5 and 6. Any amendments to the extraction template during the scoping review will be highlighted in the scoping review report.

### Analysis of the evidence

Analysis will commence with the final number of articles included in the scoping review and the basic characteristics of these articles. The data extraction template (figures 5 and 6) will highlight the most consistent and clear way for the evidence to be presented, which is representative of the extracted data once the scoping review is complete. The data extraction template will record basic characteristics of the articles' such as first author, year of publication, country, study design (ie, qualitative, quantitative, mixed-methods, retrospective, prospective, longitudinal and cohort), research methods and the number of participants in each study. This information will not be analysed but will be presented in the scoping review report to provide an overview of the included literature.

For qualitative data, content analysis will provide a description of data extracted and will be mapped to show consistencies and differences among the data.

For quantitative data, the frequency and numerical information will be presented based on the data extracted. For example, the sample size and types of study design could be individually displayed in graph or tabular format. In both the qualitative and quantitative data extraction, the analysis will reflect the objectives for the scoping review.

### Presentation of results

Results will be presented in formats such as tabular, diagrammatic and descriptive summaries to clearly represent the extracted data relevant to the scoping review objectives. The basic characteristics of the studies will be presented in table format. This will include the citation, country, study design, methods and sample size of each article. This will provide the reader with an overall summary of the studies that are included in the scoping review. Descriptive summaries of the qualitative data will be supported with visual diagrams mapping the relevant

| *Prospective or Retrospective study* | |
| --- | --- |
| *Longitudinal or Cohort study* | |
| *Research Methodology & Methods* | |
| *Participant Details (mother,father sample size)* | |
| Data extracted from sources of evidence | |
| Cancer-directed therapy options available (list) | |
| Cancer-directed therapy in conjunction with palliative care/symptoms management | YES / NO |
| Cancer-directed therapy given to child (list if more than one) | |
| Outcome from cancer-directed therapy given | |
| Values and preferences identified (list) | |
| Explanation of how these values and preferences informed decision-making (explicitly stated/not stated/unsure) | |
| Article recommendations for future research/clinical practice | |

**Figure 6** JBI data extraction tool. JBI, Joanna Briggs Institute.

categories of the scoping review objectives. For example, a word diagram to illustrate the frequency of values and preferences that come from the data and subsequently mapping these concepts into themes to show interdependent relationships between these values and preferences. Quantitative data will be presented in graphs and figures to show comparisons and differences between the data extracted. The presentation of results may be refined during the scoping review to align with the data extracted from the included articles. Amendments to the suggested presentation will be highlighted in the scoping review report.

The scoping review will identify any research gaps that could enhance how support is provided to this parent population. Recommendations for practice that may support parents when making treatment decisions for cancer-directed therapy when their child has a poor prognosis cancer will be highlighted.

## ETHICS AND DISSEMINATION

This scoping review is the first to look at parent values and preferences in treatment decision making when their child has a poor prognosis cancer. The results from this scoping review will be used to inform an interview schedule with parents to understand their values and preferences when making repeated treatment decisions for their child with relapse neuroblastoma, a poor prognosis childhood cancer. The combined results from this scoping review and parent interviews will inform the development of a decision aid to support parents in their treatment decision making. These data and the decision aid developed will be published in peer-reviewed journals and presented at national/international conferences consisting of healthcare professionals, charities and parents. This scoping review does not require ethical approval.

**Acknowledgements** This research was supported in part by theNational Institute Health Research (NIHR) Great Ormond Street Hospital Biomedical Research Centre.

**Contributors** All authors conceived the idea for this scoping review. HP developed the research questions, objectives and inclusion criteria, undertook initial database search, drafted and edited this manuscript. FG and A-SED reviewed inclusion criteria, screened abstracts and full-text papers from initial database search and edited this manuscript. All authors approved final manuscript version.

**Funding** Helen Pearson is supported by NIHR Clinical Doctoral Research Fellowship NIHR300548.

**Competing interests** None declared.

**Patient and public involvement** Patients and/or the public were not involved in the design, or conduct, or reporting, or dissemination plans of this research.

**Patient consent for publication** Not required.

**Provenance and peer review** Not commissioned; externally peer reviewed.

**ORCID iD**
Helen Pearson http://orcid.org/0000-0001-7388-3981

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
