## [Reviewer comments · BMJ Open]

ARTICLE DETAILS

TITLE (PROVISIONAL)	Parent values and preferences underpinning treatment decision-making in poor-prognosis childhood cancer: A scoping review protocol
AUTHORS	Pearson, Helen; Gibson, Faith; Darlington, Anne-Sophie

VERSION 1 – REVIEW

REVIEWER	Friedman, Danielle Memorial Sloan Kettering Cancer Center
REVIEW RETURNED	11-Dec-2020

GENERAL COMMENTS	This scoping review will provide important information about values and preferences of parents of children been treated for cancer diagnoses with poor prognosis. I do not have any concerns about the protocol and hope that the results will inform future work in this important area.
---

REVIEWER	Watson, Peter University of Cambridge, MRC Cognition and Brain Sciences Unit
REVIEW RETURNED	12-Jan-2021

GENERAL COMMENTS	Parent values and preferences underpinning treatment decision-making in poor-prognosis childhood cancer: A scoping review protocol. bmjopen-2020-046284 I would like a little more clarity on how you propose to tabulate both qualitative and quantitative outcomes and present the results, and also the choice of keywords and the choice of combinations of keywords and phrases to be used in the on-line search. Page 12, lines 8-10. My understanding is you are going to identify papers whose titles and abstracts contain specific words or phrases taken from a set list of keywords and phrases stated in Table 1 on pages 9 and 10. Characteristics of the studies in these papers will then be tabulated and/or graphed. I assume, therefore, you are graphing the FREQUENCIES of all outcomes of interest such as the number of participants and study design types (Page 12, line 19 suggests frequencies are to be used for results for quantitative data)? If this is so, for interpretability are you going to group outcomes and if so, how e.g. you could group the numbers of participants into bands of say <10, 11-50, 50-90 etc corresponding to small, medium and high sizes as is done with frequencies. Similarly how would you categorise 'study design' (line 9) e.g. single case, clinical trial, exploratory, intervention, longitudinal, cohort.
---

Page 12, line 8. I am not sure how instructive are some of the outcomes described here. I wonder, for example, how knowing the first author's name has any connection with the objective of this study of exploring parent values and preferences (Page 4, lines 6-8). How would non-specialists know anything about the study just from quoting a name like the first author? I also wonder if identifying the final author would have been more instructive as they are usually the group leader responsible for running the whole study and, as such, may be better know than the first author.

I also wonder how you would present the results of the distribution of first authors as this is qualitative? Surely with so many collaborative studies there would be a lot of authors which would make unwieldy large graphs. Would you just look at their surname or full name? Would you group author by geographic location, by university rather than present their actual (sur)name since surname may be correlated with geographical region e.g. a surname of 'Smith' may be more prevalent in English speaking countries. My main point here is what aspect of someone's name is of interest?

There are qualitative graphical methods which denote the frequency of particular words by the size of the word in a single diagram which may be useful in picturing the relative occurrences of various words. Were you thinking of using these word graphs to illustrate qualitative outcomes perhaps using colour to identify words associated with particular subgroups?

Are you looking at trends in studies over time given you are sampling over a 25 year period (page 9, line 7)?

Page 9, lines 7-10 and page 11, line 46. How many studies are you likely to identify? Would there be all that many studies given as you say on page 9, line 9 they can take years to complete and as such the resources required may not be available to many researchers. Is the third reviewer who is sampling 25% (line 46) of the studies validating the reviews of the first two reviewers? I wasn't really clear about the role of the third reviewer here. I assume the results will comprise of all the studies identified by the first two reviewers?

Page 10, Table 1. Some of the words in Table 1 strike me as so general as to be related to any area. e.g. 'uncertainty', 'decision-making' are likely not to relate specifically to cancer studies.

Page 10, line 37. Are you able to ensure you don't get duplicate articles in your searches since you are using multiple databases (Page 10, line 37) which could flag up the same paper as they would be searched separately to each other.

Page 10, line 50. You mention using a lot of 'and' and 'or' operators to search for articles of interest but I wondered how you go about deciding on optimal combinations of all the possible combinations and lengths of search phrases and keywords. With too many 'ands' you may be so specific as to miss our articles of interest whereas 'or' risks identifying too general a set of study titles which may not be related to cancer in children? I wasn't sure if you had tried to illustrate a search technique in Figure 1 (pages

	17-19) but I didn't understand what you were trying to show in this figure. Page 11, line 13. You mention contacting the corresponding author of a study for clarification but given some of these studies may go back to 1995 (page 9, line 7) would they still be contactable after such a long time? Did you think of dropout as an additional useful outcome in the longitudinal studies given interest in numbers of participants (page 12, line 9). Page 4, line 51. Should read strengThs and limitations of this study.
--	---

REVIEWER	Samuriwo, Ray Cardiff University College of Biomedical and Life Sciences, School of Healthcare Sciences
REVIEW RETURNED	18-Jan-2021

GENERAL COMMENTS	Thank you for submitting this paper on an important aspect of the care of the families of children and young persons that are diagnosed with a poor prognosis cancer. The proposed scoping review makes a novel contribution to knowledge, but there are some aspects of this paper which need to be revised as set out in the detailed comments below, i.e.: Introduction Page 6: Lines 38-42: The sentences which read “Where established treatment protocols may not be available, parents subsequently become involved in treatment decision-making including options such as clinical trials and off-protocol treatments. Often parents can make a series of repeated treatment decisions depending on their child’s response to treatment and toxicities experienced.” need to be supported with references Line 46: The sentences which reads “A poor-prognosis indicates a highly unfavorable treatment outcome likely to result in death” needs to be supported with a reference Lines 46-49: The text which reads “Emerging research has shown that 73% of parents whose child has a poor-prognosis and subsequently dies experience regret” needs to be rephrased so that the emerging research is named and referenced. This is important as point that is being made relates to the single study that is cited at the end of this sentence Lines 53-55: The text which reads “Research has acknowledged that parents pursue cancer-directed treatments even when the chance of cure is small” needs to be rephrased so that the research is named and referenced. The work cited is a decade old and it would be best to set out with greater clarity how pertinent the work cited is to contemporary practice with regards to poor-prognosis childhood cancer. Lines 55-58: The text which reads “Quality information including treatment options, risks and benefits, expectations of treatment and cure, to make informed decisions based on parent individual values and preferences is paramount” needs to be revised so that it is clear why the points made are of paramount importance. There also needs to be a definition or explanation about what can be deemed to be quality information. This should also be
--

	accompanied by some detail about who determines the quality of the information provided i.e. is it healthcare professionals, parents a combination of healthcare professionals and parents or a third party Page 7: Lines 46-48: The text which reads “The framework focuses on transition from cancer-directed therapy to palliative/End-of-Life care for their child” needs to be revised so that it is clear which framework is being referred to and this sentence needs to be underpinned with a reference Page 8 Lines 6-11: There is a typographical error in the sentence “Combining the data from this scoping review and data from parents who are making repeated treatment decisions, this will inform the development of a decision aid which is underpinned by parent values and preferences to support parents when making treatment decisions” which needs to be addressed. Lines 56-57: The text which reads “Research studies, literature or systematic reviews of qualitative and quantitative nature will be included” needs to be revised as it would be more appropriate to refer to systematic reviews of qualitative, quantitative and mixed method studies Page 9: Lines 7-9: There needs to be a more convincing rationale ideally supported with a reference to explain why the search for relevant literature is limited to the time from 1995 onward. Search strategy Page 10: Line 37: The text which reads “The databases for this review” needs to be rephrased so that it refers to the databases that will be searched. Page 11: Lines 25-28: These sentences need to be rephrased so that the initials of the two reviewers who will screen the papers as well as the third reviewer are stated. It is important to have clarity about the role that each author will play in this scoping review. Lines 30-32: The initials of the two reviewers who will read the full texts need to be stated so that there is clarity about the role of each member of the research team. Lines 45-47: There needs to be greater clarity about the role that each researcher will play in data extraction by stating the initials of each individual alongside each of the role that are set out.
--	--

VERSION 1 – AUTHOR RESPONSE

Reviewer Two Comments	Responses
Page 12, lines 8-10 My understanding is you are going to identify papers whose titles and abstracts contain specific words or phrases taken from a set list of keywords and phrases stated in Table 1. Characteristics of the studies in these papers will then be tubulated/graphed. I assume, therefore, you are graphing the frequencies of all outcomes of	At this stage it is difficult to know exactly what data will come from the scoping review. We have ideas on how this data could be presented based on other scoping reviews which have been published in the BMJ Open and what we feel is important to address the objectives of the scoping review.

interest such as the number of participants and study design types. If this is so, for interpretability are you going to group outcomes and if so, how e.g. you could group the numbers of participants into bands say <10, 11-50, 50-90 etc corresponding to small, medium and high sizes as is done with frequencies.	Frequencies related to sample size and study design have been considered. Overall this is a small parent population who are making treatment decisions for their child with a poor-prognosis cancer. The number of participants in each study is likely to be relatively small (e.g. between 5-20 participants) based on initial reading of research literature on parent decision-making in paediatric oncology. Therefore, grouping the number of participants into bands would not provide any additional clarity. A sentence has been added to 'Analysis of the evidence' section: 'For example, the sample size and types of study design could be individually displayed in graph or tabular format' Line 301-302.
Similarly how would you categorise 'study design' e.g. single case, clinical trial, exploratory, intervention, longitudinal, cohort	We have provided examples in the 'Analysis of the evidence' section with the types of categories which will be looked at in relation to study design of the included literature. These include qualitative, quantitative, mixed-methods, retrospective, prospective, longitudinal and cohort designs. Line 294-295. The data extraction template, Figure 4 has been updated to reflect this.
Page 12, line 8 I am not sure how instructive are some of the outcomes described here. I wonder, for example, how knowing the first author's name has any connection with the objective of this study exploring parent values and preferences. How would non-specialists know anything about the study just from quoting a name like the first author? I also wonder if identifying the final author would have been more instructive as they are usually the group leader responsible for running the whole study and, as such, may be better known than the first author. I also wonder how you would present the results of the distribution of first authors as this is qualitative? Surely with so many collaborative studies there would be a lot of authors which would make unwieldy large graphs. Would you just look at their surname or full name? Would you group author by geographic location, by University rather than present their actual surname since surname may be correlated with geographical region e.g. a surname of 'Smith' may be more prevalent in English speaking countries. My main point here is what aspect of someone's name is of interest?	We acknowledge that this was not completely clear, as written. We have made the following changes within the 'Analysis of the Evidence' section to make this clearer: 'The data extraction template will record basic characteristics of the articles' such as first author, year of publication, country, study design (ie: qualitative, quantitative, mixed-methods, retrospective, prospective, longitudinal, cohort), research methods, and the number of participants in each study. This information will not be analysed but will be presented in the scoping review report to provide an overview of the included literature'. Line 293-297. These characteristics will be presented in a table to show which literature was included in the scoping review. This is guidance within the JBI methodology and has been used in other published scoping reviews in BMJ Open and journals such as JBI Evidence Synthesis.

There are qualitative graphical methods which donates the frequency of particular words by the size of the word in a single diagram which may be useful in picturing the relative occurrences of various words. Were you thinking of using these word graphs to illustrate qualitative outcomes perhaps using colour to identify words associated with particular subgroups?	Again it is difficult to know at this stage exactly what data will come from the scoping review and therefore how this data could be presented. However, visual graphical methods and word graphs have been used in other published scoping review reports and is something which could support how the results are presented. The following sentence has been added to the 'Presentation of results' section: 'For example, a word diagram to illustrate the frequency of values and preferences which come from the data and subsequently mapping these concepts into themes to show inter-dependent relationships between these values and preferences'. Line 312-314.
Are you looking at trends in studies over time given you a sampling over a 25 year period?	This would be an interesting concept to look at within the data and something which could be included. It would be interesting to see whether parent value and preferences have changed over this 25 year period and if so how. Including this aspect will partly depend on the number of articles included in the scoping review to be able to get a sense of any trends over time. This is something we will consider whilst undertaking data extraction and analysis.
Page 9, lines 7-10 and page 11, line 46. How many studies are you likely to identify? Would there be all that many studies give and as you say on page 9, line 9 they can take years to complete and as such the resources required may not be available to many researchers	Due to the small population of children diagnosed with a poor-prognosis cancer and that studies take years to develop as a result, we anticipate between 10-20 studies being identified for this scoping review. This is based on our experience of knowledge in the field and initial reading on research literature relating to parent decision-making in paediatric oncology. Until the review is undertaken it is difficult to fully quantify an exact number.
Is the third reviewer who is sampling 25% of the studies validating the reviews of the first 2 reviewers? I wasn't really clear about the role of the third reviewer here. I assume the results will comprise of all the studies identified by the first two reviewers?	The following sentence has been added to the 'Data extraction' section: 'The third reviewer (FG) will review data extraction templates completed by the two full reviewers (HP and ASD) against the relevant article. This is a quality check to ensure the extracted data from the articles is accurate and is executed with rigour. Line 270-273. All identified studies will be included in the review.
Page 10, table 1. Some of the words in table 1 strike me as so general as to be related to any area. E.g. 'uncertainty', 'decision-making' are likely not to relate specifically to cancer studies	Although some of the keywords are generic and not specifically cancer related, it is how these keywords are being searched using Boolean operators that provide the focus of the scoping review topic. Table 1 has been amended for ease of reading list the keywords and associated subject headings/MeSH terms and phrases within one table.

Page 10, line 37. Are you able to ensure you don't get duplicate articles in your searches since you are using multiple databases which could flag up the same paper as they would be searched separately to each other	Searching multiple databases will lead to article duplication. The use of Endnote will enable duplicate articles to be removed. This is stated in the 'Source of evidence selection' section: 'Articles obtained will be stored on EndNote X9.2 and duplications removed'. Line 252. The PRISMA flowchart, Figure 2 provides a 'records after duplicates removed' box which will provide the reader with a clear understanding of how much duplicates were found within the searches. This will also be explicitly written within the text of the scoping review report as per the JBI methodology.
Page 10, line 50. You mentioned using a lot of 'and' and 'or' operators to search for articles of interest but I wondered how you go about deciding on optimal combinations of all the possible combinations and length of search phrases and keywords. With too many 'ands' you may be so specific as to miss our articles of interest whereas 'or' risks identifying too general a set of study titles which may not be related to cancer in children?	In preparation for this scoping review extensive reading was undertaken relating to the topic. Key articles read were used as a benchmark to ensure they were included within the database searches. These articles were used to test the search strategy to ensure optimal combinations that were not too narrow to miss relevant literature and not too wide to be unrelated to the scoping review objectives. The search technique was peer-reviewed by the University Research Librarian. To provide some idea on the number of articles yielded from the initial database searches these have been include within the 'Search strategy' as follows: 'The Medline search yielded 216 articles and the CINAHL search yielded 115 articles'. Line 207-208.
I wasn't sure if you had tried to illustrate a search technique in figure 1 but I didn't understand what you were trying to show in this figure. Note: This has now been renamed to Figure 3.	Figure 3 is showing the initial database search undertaken in Medline. Clarity of this figure has been added line 214-215: 'The Medline search strategy used in the initial database search is detailed in figure 3'. To enhance clarity we have corresponded the search strategy number within the Medline search (Figure 3) with the combination of keywords for each of the main topics to show how this search has been put together. The following sentence has been added to the 'Search strategy' section: 'The main topics combined using AND in the Medline search strategy (figure 3) are Parent (S5), Values and Preferences (S22), Decision Making (S28), Poor-Prognosis (S45), Child (S48) and Cancer (S56)'. Line 231-233.
Page 11, line 13. You mentioned contacting the corresponding author of a study for clarification but given some of these studies may go back to 1995 would they still be contactable after such a long time?	In the first instance the corresponding author will be approached if they are still working. Authors are easily searchable via online search engines, Institutional websites and platforms such as Researchgate.

	For example, seminal work in this area was conducted by Professor Hinds in 1996. Professor Hinds continues to work in this field and is still contactable. The international research collaborations within this field would provide a platform to contact the corresponding author in the first instance however if for any reason this person was not available other members of the research team would be approached. The following sentence has been added to 'Sources of evidence selection' section: 'In the absence of being able to contact the corresponding author of an article, other members of the research team will be approached'. Line 260-261.
Do you think of dropouts as an additional useful outcome in the longitudinal studies given interests in numbers of participants?	The dropout rate of longitudinal studies will be considered if this is significant in relation to the number of participants in the study. The majority of work in this field has been completed retrospectively therefore they are few longitudinal studies. To ensure we capture longitudinal data within study that might be included within the scoping review, we have added this to the data extraction template, Figure 4.
Reviewer Three Comments	Responses
Page 6, lines 38-42. The sentence which reads "Where established treatment protocols may not be available, parents subsequently become involved in treatment decision-making including options such as clinical trials and off-protocol treatments. Often parents can make a series of repeated treatment decisions depending on the child's response to treatment and toxicities experienced." need to be supported with references	This is based on authors' clinical experience. The sentence has been amended to: 'Reflections on clinical practice have identified where established treatment protocols may not be available, parents subsequently become involved in treatment decision-making. This includes options such as clinical trials and off-protocol treatments. Often parents can make a series of repeated treatment decisions depending on their child's response to treatment and toxicities experienced. This is the case for parents whose child is diagnosed with relapsed neuroblastoma or relapsed Medulloblastoma'. Line 76-81.
Page 6, line 46. The sentence which reads "A poor-prognosis indicates a highly unfavourable treatment outcome likely to result in death." needs to be supported with a reference	This sentence is now supported with a reference. Line 84.
Line 46-49: The text which reads "Emerging research has shown that 73% of parents whose child has a poor-prognosis and subsequently dies experience regret." needs to be rephrased so that the emerging research is named in referenced. This is important as the point that is being made relates to the single study that is cited at the end of this sentence	This sentence has been amended to: 'Emerging research conducted by Lichtenthal et al published in 2020 has shown that 73% of parents whose child has a poor-prognosis and subsequently dies experience regret'. Line 85.

Line 53-55: The text which reads “Research has acknowledged that parents pursue cancer directed treatments even when the chance of cure is small” needs to be rephrased so that the research is named and referenced. The work cited is a decade old and it would be best to set out with greater clarity how pertinent the work cited is to contemporary practise with regards to poor prognosis childhood cancer	This sentence has been amended to: ‘Research published in 2011 by Tomlinson et al has acknowledged that parents pursue cancer-directed treatments even when the chance of cure is small’. Line 89. Poor-prognosis childhood cancer accounts for a small patient population resulting in limited research within this field. Therefore, references reflect the latest available research within this field. The following sentence has been added: ‘Although this research is a decade old, it remains a pertinent piece of research supporting what is seen in clinical practice today. This suggests parents continue to pursue cancer-directed treatments regardless of cure outcomes’. Line 90-92.
Lines 55-58: The text which reads “Quality information including treatment options, risks and benefits, expectations of treatment and cure, to make informed decisions based on parent individual values and preferences is paramount” needs to be revised so that it is clear why the points being made are of paramount importance. There also needs to be a definition or explanation about what can be deemed to be quality information. This should also be accompanied by some detail about who determines the quality of the information provided i.e. is it healthcare professionals, parents a combination of healthcare professionals and parents or a third party	An additional sentence has been added with references: ‘This is important in treatment decision-making to increase parent empowerment, confidence in the decision made and potentially reduce decisional conflict and regret’. Line 95-96. Quality information is related to specific topics which parents consider to be important when their child is diagnosed with cancer. The following sentence has been added with a reference: ‘Quality information relating to diagnosis, treatment, expectations of cure, potential future impairments, how their child is responding to treatment and how the cancer may have developed are important factors for parents’. Line 96-98. Research has looked at parent information needs when their child has cancer. We believe the quality of this information lies in the parents’ perception of whether they have received enough information that has been explained in a coherent and understandable manner. The following sentence with references has been added: ‘Parent information needs have highlighted that good quality information results in higher levels of satisfaction in healthcare professionals and the care their child is receiving’. Line 98-100.
Page 7, lines 46-48: The text which reads “The framework focuses on transition from cancer-directed therapy to palliative/End-of-Life care for their child” needs to be revised so that it is clear which framework is being referred to and this sentence needs to be underpinned with reference	The sentence is referenced and now reads: ‘The theoretical framework is of the positive and negative influences in the decision-making process focusing on the transition from cancer-directed therapy to palliative/End-of-Life care for their child’. Line 133-134.

Page 8, lines 6-11: There is a typographical error in the sentence “Combining the data from this scoping review and data from parents who are making repeated treatment decisions, this will inform the development of a decision aid which is underpinned by parent values and preferences to support parents were making treatment decisions” which needs to be addressed	This has been divided into two sentences as follows: ‘Data from this scoping review and data from interviews with parents who are making repeated treatment decisions will be combined. This will inform the development of a decision aid which is underpinned by parent values and preferences to support parents when making treatment decisions’. Line 146-149.
Page 8, lines 56-57: The text which reads “Research studies, literature or systematic reviews of a qualitative and quantitative nature will be included” needs to be revised as it would be more appropriate to refer to systematic reviews of qualitative, quantitative and mixed methods studies.	This sentence has been amended to: ‘Qualitative, quantitative and mixed methods studies drawn from research literature including systematic reviews will be included’. Line 180-181. To clarify, this scoping review will consider all literature and not just systematic reviews.
Page 9, lines 7-9: There needs to be a more convincing rationale ideally supported with a reference to explain why the search for relevant literature is limited to the time from 1995 onwards.	The decision to start the literature search from 1995 was due to seminal work in this research field published in 1996 by Professor Hinds. The following sentence has been added with a reference: ‘This time period has been considered as seminal work was published in 1996 relating to parent treatment decision-making when their child has a first cancer recurrence’. Line 190-191.
Page 10, line 37: The text which reads “The databases for this review” needs to be rephrased so that it refers to the databases that will be searched.	This sentence has been rephrased to: ‘The databases that will be searched for this review will be.....’ Line 220.
Page 11, lines 25-28: These sentences need to be rephrased so that the initials of the two reviewers who will screen the papers as well as the third reviewer are stated. It is important to have clarity about the role that each author will play in this scoping review.	Author’s initials have been added to the following sentence: ‘This screening will be carried out independently by two reviewers (HP and ASD) before the review team meet to discuss the articles to be included for full-text review. Any discrepancies will be resolved with the addition of the third reviewer (FG)’. Lines 254-256.
Lines 30-32: The initials of the two reviewers who will read the full texts need to be stated so that there is clarity about the role of each member of the research team.	Author’s initials have been added to the following sentence: ‘These will be read in full by two reviewers (HP and ASD)’. Line 259.
Lines 45-47: There needs to be greater clarity about the role that each researcher will play in data extraction by stating the initials of each individual alongside each of the role that are set set.	Author’s initials have been added to the following sentences: ‘Data extraction will be undertaken in full by two reviewers (HP and ASD) and the third reviewer (FG) will randomly select 25% of the eligible articles to review’. Lines 269-270.

In addition, the funding statement has been updated with ‘Helen Pearson is supported by National Institute Health Research Clinical Doctoral Research Fellowship NIHR300548’. An acknowledgment section has been added with the statement ‘This research was supported in part by the NIHR Great Ormond Street Hospital Biomedical Research Centre’.

Thank you for taking the time to review our manuscript.

VERSION 2 – REVIEW

REVIEWER	Samuriwo, Ray Cardiff University College of Biomedical and Life Sciences, School of Healthcare Sciences
REVIEW RETURNED	31-Mar-2021

GENERAL COMMENTS	Thank you for submitting a revised version of the proposed scoping review on an important aspect of the care of the families of children and young persons that are diagnosed with a poor prognosis cancer. All the aspects of the original version of the manuscript that merited revision have been addressed appropriately. There are a several minor typographical errors that need to be addressed. i.e.: Abstract Page 3: There appears to be a word missing from the text which reads 'This scoping review is to explore parent values and preferences underpinning treatment decision'. I think that this sentence needs to clearly state the aim of this scoping review Introduction Page 5 Line 12-13: The sentence start of the sentence which reads 'Parent information needs' needs to rephrased to state that it is referring to research or studies that have been conducted to establish parent's information needs Search strategy Page 8 Line 54: The text which reads 'The databases that will be searched for this review will be CINAHL, Medline and PsycINFO' needs to be revised to avoid use the words will be twice in quick succession in the same sentence
--

VERSION 2 – AUTHOR RESPONSE

Reviewer Three Comments	Responses
Abstract Page 3: There appears to be a word missing from the text which reads 'This scoping review is to explore parent values and preferences underpinning treatment decision...'. I think that this sentence needs to clearly state the aim of the scoping review.	This sentence has been amended to: 'The aim of this scoping review is to explore parent values and preferences underpinning treatment decision-making when their child is receiving cancer-directed therapy for a poor-prognosis cancer.' Line 11-12.
Introduction Page 5, line 12-13: The sentence starts of the sentence which reads 'Parent information needs' needs to be rephrased to state that it is referring to research or studies that have been conducted to establish parents' information needs	This sentence has been amended to: 'Research investigating parent information needs has highlighted that good quality information results in higher levels of satisfaction in healthcare professionals and the care their child is receiving.'

	Line 87-89.
Search strategy Page 8, line 54: The text which reads 'The databases that will be searched for this review will be CINAHL, Medline and PsycINFO' needs to be revised to avoid use the words will be twice in quick succession in the same sentence	This sentence has been amended to: 'The databases to be searched for this review will be CINAHL, Medline and PsycINFO using the EBSCO platform and the Web of science core collections using the Clarivate platform.' Line 208-209.

Thank you for taking the time to review our manuscript.